# Amphotericin B-Silver Hybrid Nanoparticles Help to Unveil the Mechanism of Biological Activity of the Antibiotic: Disintegration of Cell Membranes

**DOI:** 10.3390/molecules28124687

**Published:** 2023-06-10

**Authors:** Sebastian Janik, Ewa Grela, Sylwia Stączek, Agnieszka Zdybicka-Barabas, Rafal Luchowski, Wieslaw I. Gruszecki, Wojciech Grudzinski

**Affiliations:** 1Department of Biophysics, Institute of Physics, Maria Curie-Sklodowska University, 20-031 Lublin, Poland; sebastian.janik96@gmail.com (S.J.); ewagrela123@gmail.com (E.G.); rafal.luchowski@umcs.pl (R.L.); 2Division of Biophysics, Institute of Experimental Physics, Faculty of Physics, University of Warsaw, 02-093 Warsaw, Poland; 3Department of Immunobiology, Faculty of Biology and Biotechnology, Institute of Biological Sciences, Maria Curie-Sklodowska University, 20-031 Lublin, Poland; sylwia.staczek@mail.umcs.pl (S.S.); agnieszka.zdybicka-barabas@mail.umcs.pl (A.Z.-B.)

**Keywords:** amphotericin B, nanoparticles, fungal infections, nanomedicine

## Abstract

Amphotericin B is a popular antifungal antibiotic, and despite decades of pharmacological application, the exact mode of its biological activity is still a matter of debate. Amphotericin B-silver hybrid nanoparticles (AmB-Ag) have been reported to be an extremely effective form of this antibiotic to combat fungi. Here, we analyze the interaction of AmB-Ag with *C. albicans* cells with the application of molecular spectroscopy and imaging techniques, including Raman scattering and Fluorescence Lifetime Imaging Microscopy. The results lead to the conclusion that among the main molecular mechanisms responsible for the antifungal activity of AmB is the disintegration of the cell membrane, which occurs on a timescale of minutes.

## 1. Introduction

Amphotericin B (AmB) is a life-saving antifungal antibiotic used to treat deep-seated mycotic infections [1,2] (see Appendix A). Paradoxically, despite decades of pharmacological applications of this antibiotic, the exact mode of its biological activity is still a matter of debate [3]. The formation of transmembrane channels disrupting physiological ion transport [4,5,6,7] and sequestration of membrane-bound sterols [8] are among the most frequently discussed molecular mechanisms responsible for the antibiotic properties of AmB. The significance of the challenge of understanding the molecular mechanisms responsible for the biological activity of AmB follows from the fact that this is not a purely cognitive problem. Application of the pharmaceuticals based on AmB is associated with side effects that are dramatically toxic to patients [9]. It is believed and expected that the knowledge of the mechanisms and processes underlying the antifungal effect as well as those responsible for the side effects will allow the development of a pharmacological formula and an effective treatment strategy with minimized toxicity for patients. Several pharmacological formulations of AmB have been elaborated and are used in clinical practice in order to reduce severe toxicity to patients, including a preparation containing a surfactant (e.g., Fungizone) and based on liposomes (e.g., AmBisome) [9,10]. In the present work, we address the problem of the biological activity of AmB using a system based on AmB-silver (AmB-Ag) nanoparticles previously developed in our laboratory and showing exceptionally high efficiency in combating fungal cells while being practically nontoxic to human cells at comparable concentrations [10]. The relatively high antifungal activity of AmB preparations based on metal nanoparticles was demonstrated to be associated with the fact that the antibiotic molecules in such a system can effectively cross the cellular barriers [11,12]. Particularly high antifungal activity of AmB in the form of hybrid nanoparticles with silver was originally interpreted as directly related to the effect of synergy between the antibiotic and antimicrobial activities of Ag^+^ ions [10]. On the other hand, the results of our recent studies have shown that the key mechanism in the antifungal activity of AmB is crossing the cell wall barrier, allowing direct contact of the antibiotic molecules with the cell membrane [12,13,14]. In the current work, we pose the question of whether the particularly high efficiency of AmB-Ag, previously introduced by our team [10], does not result from the unique ability of the antibiotic in the form of hybrid metallic nanoparticles to cross the cell wall barrier. Thanks to the use of molecular spectroscopy and nanoscale imaging techniques, the search for an answer to such a question can take place at the molecular level. This represents a novel approach to in-depth insight into the problem of AmB biological activity.

## 2. Results and Discussion

Figure 1 (and Appendix A) presents the FLIM (Fluorescence Lifetime Imaging Microscopy) images of *C. albicans* cells exposed to AmB-Ag nanoparticles. One of the effects of the association of AmB with silver in the form of AmB-Ag hybrid nanoparticles is the effective quenching of the fluorescence signal of the antibiotic (Figure 2). The fact that the FLIM images clearly show the short-lifetime fluorescence signal of AmB (represented by a blue color code) at the top of the long-lifetime autofluorescence of the cells (red color code) implies that the drug molecules were delivered into the cell membranes and intracellular structures in the form of AmB-Ag, but such structures were disassembled in situ so that the antibiotic molecules could effectively bind to biomembranes where the fluorescence signal of AmB is not quenched by contact with the silver nanoparticle. Such a process can be clearly observed in a model lipid membrane system. The exposition of the lipid vesicle formed with a single-lipid bilayer to AmB-Ag nanoparticles results in the binding of the antibiotic molecules to the membranes (see Figure 3). The fact that the fluorescence signal is observed in the membrane but not in the external environment, into which AmB-Ag nanoparticles were injected, implies that upon their interaction with a lipid membrane, the AmB complex with silver disassembles and the antibiotic molecules bind to a lipid environment. Moreover, the fact that the vesicle cross-section image presents a higher fluorescence signal in the right- and left-hand sectors as compared to the low- and high-membrane sectors is a direct manifestation of the fact that the direction of the transition dipole of the AmB molecule (close to the long molecular axis of AmB) is oriented vertically to the membrane plane [13]. Such an effect follows directly from photoselection since the electric vector of scanning laser light is polarized along the horizontal frame of an image [13]. The fact that the membrane-bound AmB is disassembled from the AmB-Ag structures can also be deduced on the basis of the photoselection analysis. The rationale for such a conclusion is that AmB molecules assembled randomly with the surface of silver nanoparticles would give rise to fluorescence intensity distributed homogenously around the lipid vesicle cross-section, unlike those observed in the image recorded (see Figure 3 and Appendix A). The mechanism according to which AmB-Ag nanoparticles work as a vehicle for delivering molecules of the antibiotic to various destinations within a cell can also be deduced based on the Raman scattering spectroscopy and imaging. Figure 4 presents the results of a micro-spectroscopic Raman analysis of a single *C. albicans* cell exposed to AmB-Ag. As can be seen, the resonance Raman spectrum of AmB recorded in the region of the cell membrane and in the intracellular environment are very similar to each other but different from the spectra recorded in the extracellular environment containing AmB-Ag hybrid nanoparticles. The comparison of the resonance Raman spectra of AmB unbound to metal and present in the form of nanoparticles (Appendix A) leads to the conclusion that, at last, a fraction of molecules of the antibiotic delivered to the cell are disassembled from the complex and bound directly to the structures of cells.

As can be seen from the analysis of the fluorescence images of *C. albicans* recorded after different time periods, AmB penetrates the cell almost instantly and binds to the cell membrane (Figure 1). On the other hand, a more detailed analysis of the FLIM images leads to the conclusion that a certain fraction of the antibiotic molecules binds to the cell wall, either in the form of AmB-Ag, or more likely detached from a metal nucleus. This can be deduced from the fact that the fluorescence intensity of the peripheral regions of the cell is both relatively high (not quenched by silver) and equally high, without heterogenic distribution characteristic to the AmB molecules incorporated into lipid bilayers (Figure 3 and Appendix A). Further, a detailed analysis of the distribution of fluorescence of AmB reveals that the prolonged incubation of each *C. albicans* cell (the timescale of minutes) results in the disintegration of the cell, manifested by the release of the intracellular matter labeled with the fluorescing molecules of the drug (visible as blue-colored matter in the extracellular environment). This is a clear and straightforward demonstration of the disintegrative activity of AmB, with respect to fungal cells. The association of AmB with nanoparticles made it possible to effectively deliver the antibiotic molecules into the fungal cell structures, passing the cell wall barrier shown to defend fungi against antibiotic activity [14]. This property of the hybrid nanoparticle system is most probably responsible for the exceptionally high antifungal activity of AmB-Ag [10] (see also Appendix A). Importantly, the application of this system unveiled that one of the key modes of the biological activity of AmB toward the cells of fungi is the disintegration of cell membranes and intracellular structures, based most probably on the destabilizing effect of amphiphilic molecules of this antibiotic with respect to the well-organized structure of the lipid bilayer. As demonstrated experimentally, AmB efficiently incorporates into lipid membranes that are rich in sterols (in particular ergosterol, a sterol of fungal cells) [15] and adopts a vertical orientation, with respect to the membrane plane [13]. Minimizing the energy of the system leads to the formation of aggregated AmB structures in which the polyol fragment of the antibiotic molecule is isolated from the hydrophobic core of the membrane, and the polyene fragment is exposed to it [4,16]. On the other hand, even the separation of intramembrane AmB structures from the lipid phase has a pronounced effect on the structural and dynamic properties of lipid bilayers [17,18]. It is also possible that membrane-bound AmB localizes to the protein-lipid interface and interferes with the functional organization of biomembranes. 

The results of the experiments in this work indicate a strong biological activity of AmB consisting of the disintegration of biomembranes, which, together with other modes of action of this antibiotic, leads to the death of fungal cells (see the scheme in Figure 5).

## 3. Materials and Methods

### 3.1. Materials

Amphotericin B (AmB), ergosterol (Ergo), tricine, and phosphate-buffered saline (PBS) were obtained from Merck (Darmstadt, Germany). Silver nitrate (AgNO_3_) was acquired from STANLAB (Lublin, Poland). Potassium hydroxide (KOH), potassium chloride (KCl), 2-propanol, ethanol, and chloroform were purchased from POCH (Gliwice, Poland). 1,2-dimyristoyl-sn-glycero-3-phosphocholine (DMPC) was obtained from Avanti Polar Lipids (Alabaster, AL, USA). Water used in experiments was purified by a Milli-Q Millipore system (Merck, Darmstadt, Germany).

### 3.2. Microorganism Cultivation

The yeast *Candida albicans* (wild-type; kindly gifted by Prof. A. Kędzia, Department of Oral Microbiology, Medical University of Gdansk, Gdansk, Poland) was cultivated in a YPD medium (1% yeast extract, 2% peptone, 2% glucose) at 37 °C. In the experiments, yeast in the logarithmic phase of growth was used.

### 3.3. Viability Assay

The *C. albicans* survival rate after the treatment with AmB and AmB-Ag was determined using a colony-counting assay, as described before [19]. In brief, the log-phase intact *C. albicans* cells (40 μL; OD600 = 0.3; approx. 1 × 10^3^ CFU) suspended in PBS were incubated with different concentrations of AmB or AmB-Ag (0.016–16 µg/mL) for 0.5 h at 37 °C. Control cells were incubated with 2-propanol:water 4:6 in the same conditions. After incubation, the serial dilutions were prepared and the cells were plated onto a solid YPD medium. Colonies were counted after 24 h of incubation at 37 °C. The results were calculated from three independent experiments performed in triplicate.

### 3.4. Nanoparticles Synthesis

Amphotericin B-silver hybrid nanoparticles (AmB-Ag) were synthesized according to the original procedure developed in our laboratory and described in detail in an earlier publication [10]. In the synthesis, amphotericin B acts as both a reducing agent and a stabilizing/capping agent. Thanks to alkaline synthesis, aggregation of the resulting nanoparticles was avoided. As a result, the AmB-Ag nanoparticles were monodisperse (PDI = 0.05) and had a diameter of ~7 nm, according to the results of imaging electron microscopy, dynamic light scattering, and molecular spectroscopy measurements [10]. According to the original protocol, nanoparticles were prepared with a molar ratio (AmB:Ag) of 1:11 and an antibiotic concentration of 80 µg/mL determined spectrophotometrically based on the extinction coefficient 1.3 × 10^5^ M^−1^cm^−1^ in an 2-propanol:water (4:6) solvent mixture.

### 3.5. Giant Unilamellar Vesicles (GUV) Formation

Liposomes were formed of the DMPC lipid with ergosterol at 30 mol% (with respect to lipid) by the electroformation technique as in detail described previously [20]. Lipids suspended in ethanol and chloroform were placed on two platinum electrodes. Solvents were evaporated in a vacuum for an hour. Next, electrodes were placed in the cuvette with a buffer (20 mM Tricine, 10 mM KCl, pH 7.6) for two hours of electroformation at 30 °C with an applied AC sinusoidal field (10 Hz, 3 V). 

### 3.6. Raman Spectroscopy and Imaging

Raman spectroscopy was carried out using an inVia confocal Raman microscope (Renishaw, Kingswood, UK) with an argon laser (Stellar-REN, Modu-Laser™, Centerville, UT, USA) operating at 457 nm (set at 47 μW power at the sample), equipped with a 60× water-immersed objective (Olympus Plan Apo NA = 1.2). Optical images of AmB-Ag containing GUV and *C. albicans* fungal cells were obtained and analyzed with WiRE 5.5 software (Renishaw, Kingswood, UK). Based on such images, areas of approximately 10 μm × 10 μm for Raman scanning were selected and mapped with 1 μm spatial resolution. For this study, all the images were recorded with light intensity as low as possible. Raman images were acquired using the Renishaw WiRE 5.5 system in high-resolution mapping mode (HR maps). At each point of the Raman image map, the spectra were recorded with about 1 cm^−1^ spectral resolution (2400 lines/mm grating) in Raman shift spectral region 1000–2500 cm^−1^ using an EMCCD detection camera Newton 970 from Andor, Belfast, UK. The acquisition time for a single spectrum was 0.5 s for GUV and 1.0 s for *C. albicans*. All spectra were pre-processed by cosmic ray removal, noise filtering, and baseline correction using WiRE 5.5 software from Renishaw, Kingswood, UK. 

### 3.7. Fluorescence Lifetime Imaging Microscopy (FLIM)

For analysis, 40 µL of GUV or *C. albicans* suspension was mixed with 10 µL AmB-Ag nanoparticles and imaged with the FLIM technique. Data were recorded on a MicroTime 200 confocal system (PicoQuant GmbH, Berlin, Germany) coupled with an inverted microscope (Olympus IX71, Shinjuku City, Tokyo, Japan). Silicon-immersed objective (Olympus NA = 1.3, 60×) was used during the measurements. The samples were excited with a 405 nm laser with a 10 MHz repetition rate. The observations were carried out with a 50 μm diameter pinhole, ZT 405RDC dichroic filter, and 405 nm pass filter (Chroma-AHF Analysentechnik, Tübingen, Germany). Fluorescence emission spectra were recorded with spectrograph Shamrock 163 connected to the microscopy system. Newton EMCCD DU970P BUF camera (Andor Technology, Belfast, UK) cooled to −50 °C was applied in these measurements. Fluorescence lifetimes and intensities were analyzed using SymPhoTime 64 v. 2.6 software (PiqoQuant GmbH, Berlin, Germany).

## 4. Conclusions

In this work, it was shown that the biological activity of amphotericin B, a popular antifungal antibiotic, significantly increases when the drug is in the form of hybrid silver nanoparticles. The experimental results show that AmB-Ag can effectively cross the cell wall barrier and deliver antibiotic molecules to cell membranes. Analysis of the molecular imaging data shows that AmB incorporated into the cell membrane induces membrane breakdown within minutes. Such a result clearly shows that the disintegration of fungal cell membranes is one of the main mechanisms operating at the molecular level and responsible for the biological activity of Amphotericin B.

## Figures and Tables

**Figure 1 molecules-28-04687-f001:**
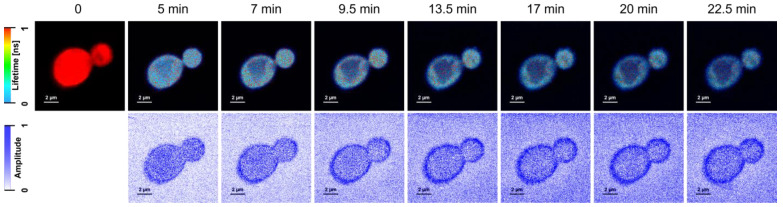
Fluorescence Lifetime Imaging Microscopy images of *C. albicans* cells before (left, 0 min) and after (time in minutes indicated) exposition to AmB-Ag nanoparticles. The upper panels present images based on fluorescence lifetime; below, the same cells are shown imaged with the based on an amplitude of the short-lifetime fluorescence component (<300 ps) assigned to AmB.

**Figure 2 molecules-28-04687-f002:**
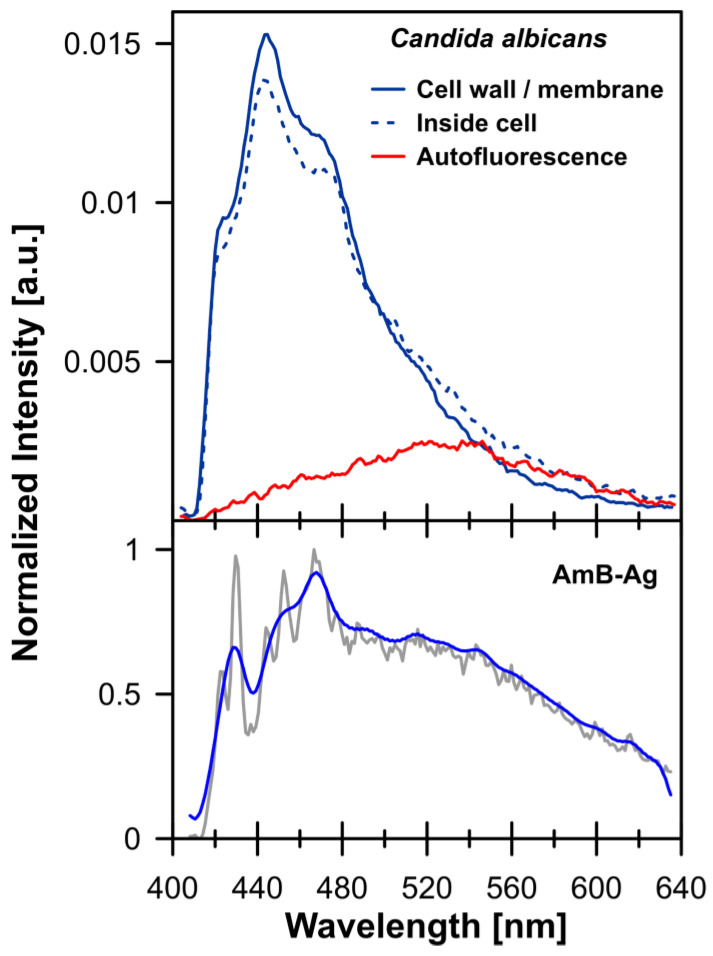
Fluorescence emission spectra recorded from individual areas of the *C. albicans* cell (cell wall/membrane: solid blue line and inside cell area: dotted blue line) and autofluorescence signal: solid red line (upper panel). Spectra of amphotericin B were normalized by the area under the curve, and the autofluorescence emission spectrum was normalized in the region where there is no amphotericin B emission: around 620 nm. In the lower panel, fluorescence emission spectra of AmB-Ag nanoparticles are shown: raw spectrum (solid blue line) and smoothed spectrum (solid gray line). The relatively low signal-to-noise ratio observed in the case of the AmB-Ag sample represents the fact of highly effective fluorescence quenching by the metal surface of a nanoparticle.

**Figure 3 molecules-28-04687-f003:**
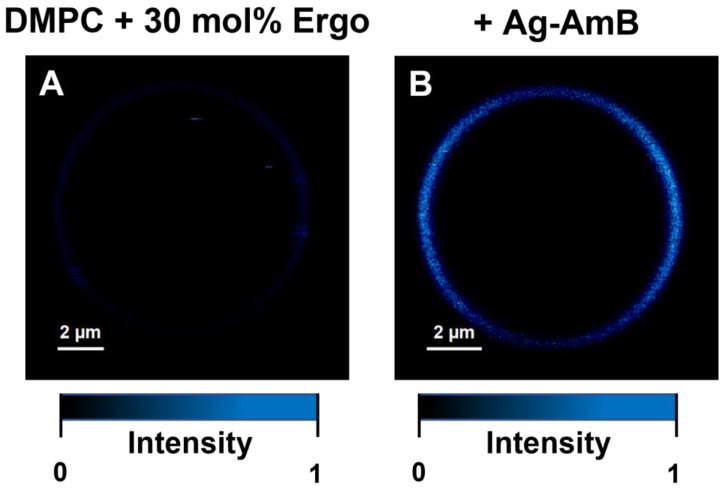
Results of microscopic imaging of single-lipid vesicles composed of DMPC with 30 mol% Ergo (**left panel**) and with the addition of amphotericin B (**right panel**). The images are based on fluorescence intensity. The images represent an equatorial vesicle cross-section in the focal plane of the microscope. Maximum fluorescence emission intensities (displayed in blue) on the left and right sides of the liposome represent the fraction of AmB molecules incorporated perpendicular to the membrane plane (vertical orientation).

**Figure 4 molecules-28-04687-f004:**
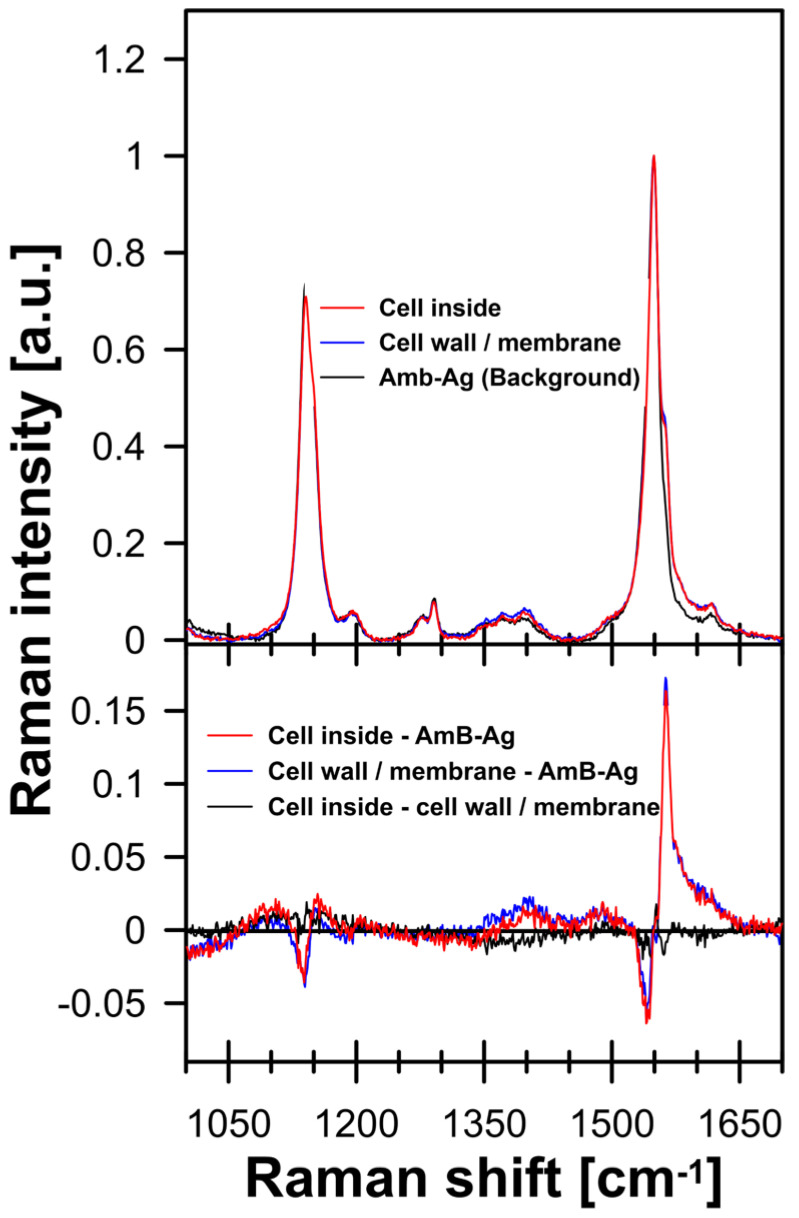
Raman spectra of AmB recorded with the application of a microscopic system from different compartments of *C. albicans* cell. Spectra were recorded with 458 nm laser line. Laser excitation and detection of the Raman scattering signal were focused inside the cell (red line), outside the cell (black line), and in the cell wall/cell membrane region (blue line). The lower panel shows the difference spectra calculated based on the original spectra displayed in the upper panel (see the legend).

**Figure 5 molecules-28-04687-f005:**
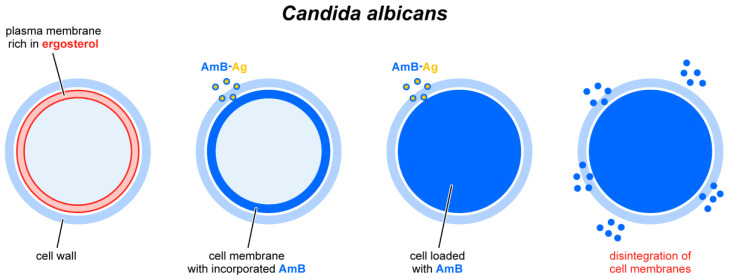
Schematic drawing showing the successive stages of interaction of AmB-Ag nanoparticles with a cell: crossing the cell wall barrier and integration with the cell membrane, penetration into the cytoplasm, disintegration of the cell membrane, and outflow of cell content to the external environment.

## Data Availability

Not applicable.

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
