# Peer review of "Amphotericin B-Silver Hybrid Nanoparticles Help to Unveil the Mechanism of Biological Activity of the Antibiotic: Disintegration of Cell Membranes"

_molecules, 2023, doi:10.3390/molecules28124687_

Round 1

Reviewer 1 Report

Please find my comments below:

1) The abstract has to be substantially improved completely.

2) The authors have not stated the novelty of this work at the end of Introduction section. 

3) The authors must discuss several other previously reported studies in the Introduction part.

4) The conclusion must be expanded. 

5) The English language has to be substantially improved throughout the manuscript. Please use English language editing service. 

6) Where is MIC value?

7) What is the particle size and zeta potential of the NPs?

8) Where is SEM and TEM images?

9) Why thorough characterization for NPs has not been reported? It is mandatory.

English language has to be substantially improved.

Author Response

Answers to the points raised by the Reviewers

We would like to thank the reviewers for their valuable comments and suggestions which have helped us to prepare a revised version of our paper.

The answers to the points raised by the reviewers are placed below along with the description of the changes made.

Reviewer #1

  • The abstract has to be substantially improved completely.

Answer:

The abstract  has been rewritten

  • The authors have not stated the novelty of this work at the end of Introduction section. 

Answer:

A novelty of the present work has been emphasized in the Introduction section of the corrected version of our manuscript.

  • The authors must discuss several other previously reported studies in the Introduction part.

Answer:

The Introduction section has been rewritten.

  • The conclusion must be expanded. 

Answer:

The conclusion has been expanded, as suggested.

  • The English language has to be substantially improved throughout the manuscript. Please use English language editing service.

Answer:

The English has been checked and corrected.

  • Where is MIC value?

Answer:

The MIC values for our system were determined and described in our previous work (Tutaj et al. Nanomedicine: Nanotechnology, Biology, and Medicine 12 (2016) 1095–1103.). We refer to this in the revised version of this manuscript.

  • What is the particle size and zeta potential of the NPs?

Answer:

The particle size was analyzed and reported in our previous publication in which the system has been precisely described and characterized (Tutaj et al. Nanomedicine: Nanotechnology, Biology, and Medicine 12 (2016) 1095–1103.). In the revised version of our paper, we report the sizes and refer more specifically to this publication.

  • Where is SEM and TEM images?

Answer:

The original images were reported in our previous publication in which the system has been precisely described and characterized (Tutaj et al. Nanomedicine: Nanotechnology, Biology, and Medicine 12 (2016) 1095–1103.). In the revised version of our paper, we refer more specifically and directly to this publication.

  • Why thorough characterization for NPs has not been reported? It is mandatory.

Answer:

Our original system based on hybrid amphotericin B-silver nanoparticles was introduced and thoroughly characterized in one of our earlier works (Tutaj et al. Nanomedicine: Nanotechnology, Biology, and Medicine 12 (2016) 1095–1103.). The original paper referred to a large number of supplementary materials in which the system was extensively characterized. In the revised version of this article, we provide additional information about our system and refer more specifically and directly to our previous work.

Reviewer 2 Report

This study investigates interaction of hybrid Amphotericin B-Ag nanoparticles with C. albicans cells. With the help of molecular imaging techniques based on time-resolved fluorescence spectroscopy and Raman scattering, the authors have deduced a mechanism of antifungal action of the hybrid. The manuscript is not suitable for publication in its current state many things need to be modified and the most important notes are:

The information provided in the article is too less, if it was submitted as a communication then it is acceptable, but for a full article, a full background of the study with corresponding literature, details of the material preparation (such as the preparation of hybrid), in-depth characterization and more detailed discussion about the results are required.

The title is confusing (no where it was mentioned in the title that the study is about the hybrid of Ag NPs with the drug). Therefore, it has to be revised

The abstract is very week, needs to revised by including more clear results and novelty of the work

The introduction is very short, with many unclear sentences with grammar and spelling mistakes, thus it needs extensive editing.

see the report

Author Response

Answers to the points raised by the Reviewers

We would like to thank the reviewers for their valuable comments and suggestions which have helped us to prepare a revised version of our paper.

The answers to the points raised by the reviewers are placed below along with the description of the changes made.

Reviewer #2

  1. The information provided in the article is too less, if it was submitted as a communication then it is acceptable, but for a full article, a full background of the study with corresponding literature, details of the material preparation (such as the preparation of hybrid), in-depth characterization and more detailed discussion about the results are required.

Answer:

Since the present paper is a continuation of our previous work we decided not to provide all the experimental results and instead to refer to our original publication in where the hybrid Amphotericin B-silver nanoparticle system has been precisely described and characterized (Tutaj et al. Nanomedicine: Nanotechnology, Biology, and Medicine 12 (2016) 1095–1103.) On the other hand, we fully agree with the Reviewer that more information on the experimental system should be provided in the present article. The experimental section has been expanded accordingly.

  1. The title is confusing (no where it was mentioned in the title that the study is about the hybrid of Ag NPs with the drug). Therefore, it has to be revised

Answer:

We thank the reviewer for this suggestion. The article title has been modified.

  1. The abstract is very week, needs to revised by including more clear results and novelty of the work

Answer:

The abstract has been rewritten.

  1. The introduction is very short, with many unclear sentences with grammar and spelling mistakes, thus it needs extensive editing.

Answer:

The Introduction section has been modified.

Round 2

Reviewer 1 Report

The authors have clarified my concerns. 

Minor English language editing is required.

Reviewer 2 Report

The authors have addresses all the comments raised by the reviewer, not the manuscript is suitable to be accepted

minor spell checking required